# Prevention of Melanoma Extravasation as a New Treatment Option Exemplified by p38/MK2 Inhibition

**DOI:** 10.3390/ijms21218344

**Published:** 2020-11-06

**Authors:** Peter Petzelbauer

**Affiliations:** Skin & Endothelial Research Division, Department of Dermatology, Medical University Vienna, A-1090 Vienna, Austria; Peter.Petzelbauer@meduniwien.ac.at

**Keywords:** melanoma, extravasation, p38, MK2, endothelium

## Abstract

Melanoma releases numerous tumor cells into the circulation; however, only a very small fraction of these cells is able to establish distant metastasis. Intravascular survival of circulating tumor cells is limited through hemodynamic forces and by the lack of matrix interactions. The extravasation step is, thus, of unique importance to establish metastasis. Similar to leukocyte extravasation, this process is under the control of adhesion molecule pairs expressed on melanoma and endothelial cells, and as for leukocytes, ligands need to be adequately presented on cell surfaces. Based on melanoma plasticity, there is considerable heterogeneity even within one tumor and one patient resulting in a mixture of invasive or proliferative cells. The molecular control for this switch is still ill-defined. Recently, the balance between two kinase pathways, p38 and JNK, has been shown to determine growth characteristics of melanoma. While an active JNK pathway induces a proliferative phenotype with reduced invasive features, an active p38/MK2 pathway results in an invasive phenotype and supports the extravasation step via the expression of molecules capable of binding to endothelial integrins. Therapeutic targeting of MK2 to prevent extravasation might reduce metastatic spread.

## 1. Melanoma Dissemination

Melanoma arises from melanocytes residing in the basal layers of the epidermis or from nevi. In the initial phase, melanoma cells spread horizontally within the epidermis. At variable intervals, these cells vertically invade into the dermis. For vascular entry, cancer cells may directly enter tumor blood vessels, which are in a state of continuous reconfiguration resulting in a tortuous, leaky and permeable phenotype [1,2]. However, primary melanoma mainly disseminates by the lymphatic route to reach the sentinel lymph node [3]. Mechanisms guiding lymphatic entry are still poorly characterized. Tumor cells may use channels between button-like intercellular junctions of lymphatic capillaries or by actively producing holes between endothelial cells [4,5]. Lymphatic entry is then thought to be based on chemokine gradients rather than on active adhesion to lymphatic endothelium [6], which is in contrast to interactions with blood vessel endothelium. Through further lymphatic drainage, they may reach the blood circulation. Within the blood stream, circulating tumor cells are in an intermediate state of the metastatic cascade. Tumors may release numerous tumor cells into the circulation; however, mostly only a very small fraction of these cells is able to establish distant metastasis [7]. Intravascular survival of circulating tumor cells depends on their resistance to destructive hemodynamic forces, which requires mechano-adaption [8]. More importantly, anchorage-dependent cells when detached from the surrounding extracellular matrix undergo anoikis, a form of programmed cell death. Tumor cells may use various mechanisms to escape anoikis, such as interaction with platelets or myeloid cells, or by activation of signaling pathways, such as Wnt or STAT3 [9,10,11]. In case of intravascular survival followed by successful extravasation, effective colonization of distant organs or tumor dormancy at this site depends on numerous factors, e.g., the angiogenic switch, immune escape, matrix remodeling [12].

Therapeutic approaches could theoretically tackle each step, although current therapeutic strategies aim at reducing melanoma growth and/or reducing total tumor load. This can be achieved by blocking overactive kinase signaling in melanoma cells by currently targeting the Ras/Raf/MEK/ERK pathways. A more recent and more successful approach aimed at inducing an immune response by blocking proteins with immune-inhibitory functions, such as CTLA-4 or PD-1, PD-L1 in order to foster immune-cell-mediated tumor cell killing. In spite of these advances, melanoma is still associated with significant mortality requiring additional therapeutic approaches. Prevention of extravasation of melanoma cells from the blood stream into tissues has not entered the clinic as a supportive strategy to reduce metastatic spread.

## 2. Melanoma Extravasation

Extravasation of cancer cells is an active process; this was first observed by in vivo imaging approximately 60 years ago. The cell size is not the determining factor in the incidence of intravascular arrest, and tumor cells are not entrapped in capillaries with a smaller diameter than cancer cells. They actively adhere to the endothelium of vessels with a diameter wider than that of the tumor cells [13,14]. This suggests that factors beyond size restriction are involved in tumor cell arrest, leading to the concept that circulating tumor cells must become “sticky” to the luminal surface of blood vessels.

The view that circulating cells have to actively stick to endothelial surfaces in order to extravasate was experimentally substantiated more than 40 years ago by demonstrating that inflammatory cells roll on endothelial surface molecules before being able to stick and eventually extravasate [15]. Since then, numerous molecules of the selectin, immunoglobulin superfamily receptors, junctional proteins and integrins have been investigated in their efficacy to mediate interactions between leukocytes and endothelium in order to marginalize circulating leukocytes to the luminal surface of blood vessels, to mediate their adhesion to endothelium and support their extravasation [16]. Whereas for leukocyte extravasation, the molecules involved and the hierarchy of events are now relatively well understood, the mechanisms involved in extravasation of cancer cells are still only partially understood and even less information is available for melanoma.

Among several reasons, the experimental setup seems to be a major stumbling block. If metastasis in spontaneous tumor models is used as an endpoint, this reflects the sum of events as described above but does not give information on the efficiency of the extravasation step. Intravital microscopy as done for the dissection of leukocyte adhesion/extravasation is hampered by the relatively rare event of circulating tumor cells in spontaneous tumor models [17]. Even in the “simple” model of intravenous injection of tumor cells, intravital microscopy to monitor extravasation is limited by the selection and size of the operating window. Moreover, in the intravenous injection model, in addition to the quantification of tumor cells reaching the tissues, ideally concomitantly intravascular tumor cells should be monitored for anoikis, which is very rarely done. Thus, most of the information on endothelial/melanoma cell interactions stems from cell culture models.

### 2.1. The Rolling and Adhesion Step

Selectins expressed on endothelial cells mediate rolling of circulating cancer cells through interaction with sialyl Lewis X-containing glycoproteins [18]. However, on melanoma, the expression of these carbohydrate moieties is not well characterized. As shown for sialyl-Lewis(x/a)-negative melanoma cells, its function may be replaced by interactions of integrin α4β1 and endothelial vascular adhesion molecule (VCAM-1) mediating melanoma adhesion and subsequent extravasation under conditions of flow [19]. The importance of VLA-4 expressed on melanoma cells that mediates adhesion to endothelial VCAM-1 to promote extravasation is also shown in mouse models following intravenous injection of melanoma cells [20,21]. Melanoma cells also target endothelial VCAM-1 through tumor-derived SPARC, which induces paracellular endothelial permeability. Blocking VCAM-1 impedes melanoma-induced endothelial permeability in vitro and lung metastasis in a mouse model following intravenous injection [22].

### 2.2. The Extravasation Step

The extravasation step, i.e., diapedesis, requires opening of endothelial cell-cell junctions. For leukocytes, the interaction with junctional adhesion molecules JAM-A, B and C and the distantly related ESAM is relevant [23]. JAM-C, but not A and B, can be found expressed in human melanoma. JAM-C is able to undergo homophilic binding or heterophilic interactions with the leukocyte integrin Mac-1 or JAM-B. In in vitro models, JAM-C was shown to mediate transmigration of melanoma cells through endothelial cells. JAM-C blockade in vivo prevented lung metastasis in a murine model following intravenous injection [24]. Using similar experimental approaches as described above, CD146 expressed on tumor and on endothelial cells was shown to promote tumor cell extravasation and metastasis through homophilic interactions [25,26]. Compared to the amount of molecules that are known to be involved in leukocyte diapedesis [23], the knowledge on their role in melanoma diapedesis is limited, although many of these molecules are expressed in melanoma cells such as CD99, CD99L2 and CD54.

## 3. Phenotypic Plasticity of Melanoma

In contrast to physiological cells where the transition from a stem cell to a differentiated cell tends to be unidirectional, in cancer, transitions between different phenotypic states appear dynamic and potentially reversible. As recently reviewed, specifically for melanoma, well-defined biomarkers for distinct melanoma cellular phenotypic states exist that characterize growth behavior and invasive potential, and these different phenotypes may have coexisted within one tumor in vivo [27]. For example, heterogeneity within one tumor was demonstrated when using JARID1B as a biomarker. Expression of JARID1B is dynamically regulated and defines a small subpopulation of slow-cycling melanoma cells, but JARID1B-negative cells can become positive, and even single melanoma cells irrespective of selection are tumorigenic [28]. Another approach using transcriptional profiling identified the melanocyte master transcriptional activator gene MITF and the receptor tyrosine kinase gene AXL to describe different cellular states, i.e., between proliferation versus invasion. Within individual tumors, MITF^high^-AXL^low^ cells have a proliferative phenotype and MITF^low^-AXL^high^ cells are invasive [29]. Signaling pathways activated downstream of AXL include PI3K-AKT-mTOR, MEK-ERK, NF-κB, and JAK/STAT [30]. Another study by Widmer et al. characterized a set of 98 genes which discriminate proliferative from invasive cells [31]. Expression of this set of genes does not depend on the BRAF^V600E^ mutation [31,32]. Genes expressed in MITF^high^ cells [29] show a considerable overlap with proliferative genes described by Widmer et al., whereas genes expressed in AXL^high^ cells are distinct from invasive genes as defined by Widmer et al. (3 out of 45 overlap with genes expressed in AXL^high^ cells). Thus, the transcriptional program associated with an invasive phenotype of melanoma is still imprecise and leaves unanswered the question of which signaling pathways are involved in the expression of molecules supporting melanoma cell adhesion and extravasation.

### The MAP Kinase Pathways

MAPK pathways have repeatedly been discussed in melanoma progression. The role of the RAF/MEK/ERK pathway is best characterized because of the high frequency of BRAF mutations [33]. However, neither the phenotype invasive versus proliferative nor levels of AXL expression correlate with BRAF^V600E^ mutation [31,34]. In regard to p38 and JNK, the mutual interference between these two pathways has been established [35]. Very little is known about the role of JNK in melanoma progression. Most knowledge comes from investigations analyzing the function of targets of JNK—mainly AP-1 proteins—but not JNK itself. The JNK/AP1 axis promotes melanoma cell proliferation [36]. Blocking JNK leads to p38 activation, and p38 signaling in melanoma results in highly mobile and lymphangiogenic melanoma cells expressing low levels of MITF, suggesting that p38 induces an invasive phenotype [37].

The p38 MAPK family consists of four different kinases: p38α (MAPK14), p38β (MAPK11), p38γ (MAPK12) and p38δ (MAPK13). p38α and p38β share a sequence homology of 75% and are ubiquitously expressed. Except for p38δ, all other isoforms are expressed in melanoma [38]. p38 MAPKs are activated in response to stress signals like pro-inflammatory cytokines, heat shock, UV light, ischemia and hypoxia [38]. Activation is mediated by the MAPK signaling module, whereas autoactivation is mediated by the TGFβ-activated kinase 1 binding protein 1 (TAB1) [39] or by T cell receptor-proximal tyrosine kinases [40]. Within the MAPK signaling module, MKK3 and MKK6 share 82% homology specific to p38 activation. The p38 MAPK family activates more than 100 substrates by direct phosphorylation. Most downstream targets are transcription factors (e.g., p53, ATF2, ELK1, MEF2, C/EBPβ), while others are protein kinases (e.g., MK2, MSK1, MNK1, MNK2), phosphatases, cell-cycle/apoptosis regulators, growth factor receptors and cytoskeletal proteins [41]. Moreover, several direct or indirect p38 targets are involved in melanoma progression, such as fibronectin, Wnt-5A or VEGF-C [37,42,43]. Silencing p38α in melanoma downregulates matrix metallopeptidases 2 and 9, TWIST1, Zinc finger protein SNAI1, VEGF-C and vimentin. It upregulates E-cadherin which is suggestive of the induction of a proliferative phenotype [44]. Interestingly, transgenic expression of CDH1 in melanoma blocks p38 signaling, suggesting that the loss of CDH1 leads to a more active p38 pathway indicating a direct crosstalk between p38 activity and CDH1 expression [45]. A detailed summary of the regulation of E-cadherin expression is shown in Figure 1. Analyzing gene expression in 22 patient-derived melanoma lines confirmed the role of p38 signaling in phenotype switching of melanoma from proliferative to invasive [46]. Moreover, this study identified the mitogen-activated protein kinase-activated protein kinase 2 (MAPKAPK2 or MK2) as the relevant downstream target of p38 in mediating these effects. Signaling through p38/MK2 causes a loss of CDH1 expression and induces an invasive phenotype. Blocking MK2 restores CDH1 expression. In vivo, in an intravenous model, blocking MK2 activity reduces lung metastases, implying an impaired ability to extravasate.

## 4. Melanoma/Endothelial Interaction

The interaction of melanoma cells with endothelium can be analyzed in in vitro co-culture models. For example, electrical resistance across endothelial monolayers is a surrogate for the integrity of endothelial cell–cell junctions. Melanoma cells added to endothelial monolayers reduced electrical resistance suggestive of junction disruption. Inhibition of MK2 in melanoma cells reduced the ability of melanoma to reduce endothelial electrical resistance [46]. In contrast to VEGF-A, which disrupted endothelial junctions for very short time intervals, endothelial disruption caused by melanoma cells was long-lasting, suggesting that the melanoma-induced decrease in endothelial resistance is due to endothelial cell retraction [46,55]. This was confirmed in another assay, where melanoma spheroids, seeded on endothelial layers, induced endothelial retraction limited to the contact points of spheroids, which was significantly reduced by MK2 inhibition. A negative correlation analysis of CDH1 expression in relation to the proteins that activate integrins through RGD sequences, which potently induce endothelial retraction, revealed DEL-1 (developmental endothelial locus-1, also called EDIL3), PODXL (podocalyxin-like protein 1) and THBS1 (thrombospondin 1) in melanoma cells as p38/MK2 targets that mediate melanoma invasiveness [46]. Details of the regulatory pathway are shown in Figure 2. The Cancer Genome Atlas Skin Cutaneous Melanoma analysis showed that the high expression of DEL-1 and PODXL/mRNA in samples from patients with melanoma was indeed associated with poor survival and could be used as a prognostic marker [46].

## 5. Therapeutic Outlook

Preclinical data favor the use of small molecule p38 kinase inhibitors in patients, but none of the clinical trials have progressed to Phase III mainly due to severe side effects such as cardiac toxicity, hepatotoxicity and the induction of central nervous system disorders. MK2—a direct target of p38—might be a better target for cancer treatment than the poly-functional p38. This is supported by experiments with p38α knockout mice that are embryonic lethal, whereas viability of MK2-knockout mice is not affected [62]. MK2 has a proline-rich N-terminal region that is connected to the kinase catalytic domain containing two major phosphorylation sites (Thr222 and Ser272). A third phosphorylation site (Thr334) precedes the nuclear export signal and a nuclear localization signal. The C-terminal domain contains the p38-binding site and an ATP-binding site [63,64]. Most MK2 inhibitors are ATP-competitive MK2 inhibitors competing with intracellular ATP molecules to block p38MAPK-mediated phosphorylation and activation of MK2 but suffer from low solubility, poor cell permeability and poor kinase selectivity. Non-ATP-competitive inhibitors interact with a binding site in the kinase, which is different from that of ATP, thus avoiding issues such as selectivity with other kinases. An additional advantage is their effectiveness at low concentration [65]. However, the biological significance of MK2 in skin tumor progression in clinical studies is not well elucidated. For the development of squamous cell carcinoma, MK2-deficient mice develop fewer skin carcinomas as compared to wild-type mice when induced by initiation with 7,12-dimethylbenz[a]anthracene and 12-O-tetradecanoylphorbol-13-acetate [66]. Several preclinical studies have suggested that MK2 inhibition may also be protective in several types of cancer as recently reviewed [67]. The potential impact of combining MK-2 inhibition with inhibitors of BRAF or MEK is yet difficult to predict; experiments in *KRAS*-mutant tumor cells would suggest a synergistic effect [68].

## 6. Conclusions

Melanoma metastasis and not the primary tumor is the cause of death of the patient. While most therapies aim at the inhibition of tumor growth, there are no approved drugs targeting the extravasation step required for successful tumor dissemination. Similar to leukocytes, melanoma cells require multiple proteins to interact with endothelium, each offering possibilities for therapeutic intervention. This perspective emphasizes the role of MK2 signaling in melanoma for successful interactions with blood vessel endothelium. MK2 inhibitors are available and thus offer themselves as an adjuvant treatment to prevent/reduce melanoma extravasation and metastatic spread.

## Figures and Tables

**Figure 1 ijms-21-08344-f001:**
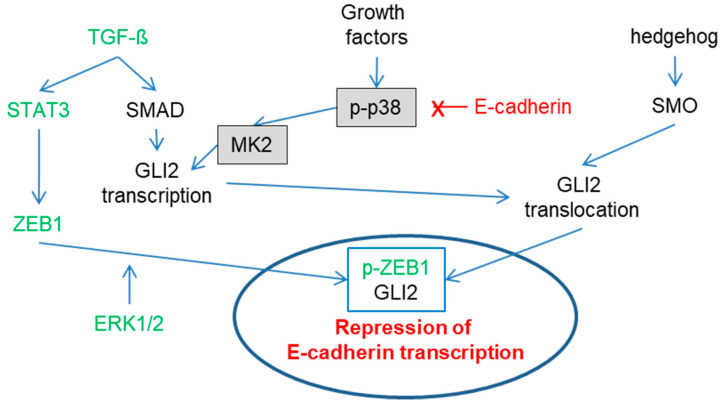
Complex regulation of E-cadherin expression. For ZEB1: STAT3 induces zinc finger E-box-binding homeobox 1 (ZEB1) transcription [47], and ZEB1 is then phosphorylated by ERK1/2 [48] followed by nuclear localization. For GLI2: TGF-ß via SMAD and p38 via MK2 induce GLI2 transcription [45,49,50]. Activation of SMO through the hedgehog pathway leads to GLI2 nuclear translocation [51,52]. Nuclear ZEB1 and GLI2 repress E-cadherin expression [53]. Interestingly, E-cadherin (X) expression reduces activation of the p38 signaling pathway [45], potentially via nuclear localization [54], thereby potentially counteracting its own downregulation.

**Figure 2 ijms-21-08344-f002:**
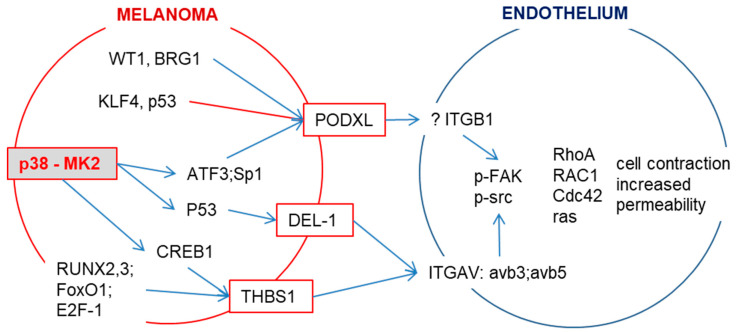
p38/MK2 signaling induces the expression of integrin-binding proteins. p38/MK2 signaling in melanoma cells induces the expression of PODXL (podocalyxin-like protein 1), DEL-1 (developmental endothelial locus-1) and THBS1 (thrombospondin 1), which can interact with endothelial integrins (ITGB1 or ITGAV). This interaction results in the phosphorylation of FAK and Src followed by the activation of GTPases. This induces stress fiber formation and puts force on endothelial junctions, thereby regulating permeability. DEL-1 is a secreted protein and a prognostic factor for overall survival in carcinoma and melanoma patients [46,56]. PODXL, a member of the CD34 family of transmembrane sialomucins, is a marker for poor prognosis in melanoma and several types of cancer [46,57,58,59]. PODXL itself does not contain a Arg-Gly-Asp motif; it indirectly enhances the adherence of cells to endothelium through RGD-dependent activation of integrins [60]. THBS1 expression predicts poor prognosis in breast and gynecological cancers [61].

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
