# Peer review of "Prevention of Melanoma Extravasation as a New Treatment Option Exemplified by p38/MK2 Inhibition"

_ijms, 2020, doi:10.3390/ijms21218344_

Round 1

Reviewer 1 Report

In the present perspective “Prevention of Melanoma Extravasation as a new Treatment Option Exemplified by p38/MK2 Inhibition”, the author discusses the role of p38/MK2 signaling in melanoma extravasation/ vascular dissemination/ metastasis and elucidates its interaction/ crosstalk with endothelial integrins. In conclusion, the author provides a promising therapeutic target to prevent/reduce melanoma extravasation and metastatic spread that eventually leads to death of melanoma patients.

In general, this manuscript is very well-written and presents interesting insights regarding the melanoma – endothelial interaction elucidating its pathways and various markers, thus adding more data in this field of research.

I have some minor suggestions/ typo corrections.

(lane 124) “Signaling pathways activated downstream of AXL include PI3K-AKT-mTOR, MEKERK, NF-κB, and JAK/STAT ……Please add “dash” so that it is MEK-ERK

(lane 157) It upregulates E-cadherin which is suggestive for the induction of an invasive phenotype ……Please correct the typo as “suggestive of”

(lane 173) Nuclear ZEB1 and 173 Gli2 repress E-cadherin expression ….Please correct typo GLI2.

Author Response

I thank the reviewer very much for the positive comments and I have made all the suggested corrections.

(lane 124) “Signaling pathways activated downstream of AXL include PI3K-AKT-mTOR, MEKERK, NF-κB, and JAK/STAT ……Please add “dash” so that it is MEK-ERK

(lane 157) It upregulates E-cadherin which is suggestive for the induction of an invasive phenotype ……Please correct the typo as “suggestive of”

(lane 173) Nuclear ZEB1 and 173 Gli2 repress E-cadherin expression ….Please correct typo GLI2.

Reviewer 2 Report

I appreciate the effort of Dr. Petzelbauer to describe the complex pathways regulating the melanoma extravasation pointing out the phenotypic plasticity of melanoma. The topic of his perspective is very interesting, but the paper is too difficult to read and some sentences are convoluted and the meaning is not clear.

For instance:

Lines 112-114: “Most of the experiments investigating melanoma/endothelial interactions are done with melanoma cell lines not reflecting the fact that a static, hierarchical cancer stem cell model cannot be applied to understand melanoma biology.”  

The two models (dynamic plasticity or hierarchical CSC) are still debated for melanoma

Line 115: “…individual melanoma cells are capable of switching between cellular states.” Here, it should be specify to which cellular states this assessment refers.

Lines 156-158: “Silencing p38α in melanoma downregulates matrix metallopeptidases 2 & 9, TWIST1, Zinc finger protein SNAI1, VEGF-C and vimentin. It upregulates E-cadherin which is suggestive for the induction of an invasive phenotype”.  It seems that high E-cadherin induces an invasive phenotype that it is not correct.

Lines 178-179: “In in vitro models, MK2 inhibition reduced the melanoma-induced reduction of endothelial electrical resistance as a surrogate for the integrity of endothelial cell-cell junctions”. Not clear

Moreover, to facilitate the comprehension, I suggest to revise the structure of paragraphs taking into account that “The Rolling & Adhesion Step” and “The extravasation step” sections are subsections of the “Melanoma Extravasation” section, then they might be indicated as 2.1 and 2.2 sections respectively. Also “The MAP kinase pathways” section should be indicated as subsection (3.1) of the “Phenotypic plasticity of melanoma” section.

Therefore, the paper needs to be accurately reviewed to clarify the important concepts that are reported.

Author Response

I thank the reviewer very much for the positive comments and I have made all the suggested corrections.

Lines 112-114: “Most of the experiments investigating melanoma/endothelial interactions are done with melanoma cell lines not reflecting the fact that a static, hierarchical cancer stem cell model cannot be applied to understand melanoma biology.” The two models (dynamic plasticity or hierarchical CSC) are still debated for melanoma.

Response: This sentence is now deleted and the paragraph reads now as follows:

  1. Phenotypic plasticity of melanoma

In contrast to physiological cells where the transition from a stem cell to a differentiated cell tends to be unidirectional, in cancer, transitions between different phenotypic states appear dynamic and potentially reversible. As recently reviewed, specifically for melanoma, well-defined biomarkers for distinct melanoma cellular phenotypic states exist that characterize growth behavior and invasive potential and this different phenotypes may even coexisted within one tumor in vivo [27]. For example, heterogeneity within one tumor was demonstrated,

Line 115: “…individual melanoma cells are capable of switching between cellular states.” Here, it should be specify to which cellular states this assessment refers.

Response: i.e., between proliferation versus invasion, this is now included.

Lines 156-158: “Silencing p38α in melanoma downregulates matrix metallopeptidases 2 & 9, TWIST1, Zinc finger protein SNAI1, VEGF-C and vimentin. It upregulates E-cadherin which is suggestive for the induction of an invasive phenotype”.  It seems that high E-cadherin induces an invasive phenotype that it is not correct.

Response: Thank you for this comment - this typo is now corrected to: …..induction of a proliferative phenotype.

Lines 178-179: “In in vitro models, MK2 inhibition reduced the melanoma-induced reduction of endothelial electrical resistance as a surrogate for the integrity of endothelial cell-cell junctions”. Not clear

Response: This is reworded as follows: The interaction of melanoma cells with endothelium can be analyzed in in vitro co-culture models. For example, electrical resistance across endothelial monolayers is a surrogate for the integrity of endothelial cell-cell junctions. Melanoma cells added to endothelial monolayers reduced electrical resistance suggestive for junction disruption. Inhibition of MK2 in melanoma cells reduced the ability of melanoma to reduce endothelial electrical resistance [45].

Moreover, to facilitate the comprehension, I suggest to revise the structure of paragraphs taking into account that “The Rolling & Adhesion Step” and “The extravasation step” sections are subsections of the “Melanoma Extravasation” section, then they might be indicated as 2.1 and 2.2 sections respectively. Also “The MAP kinase pathways” section should be indicated as subsection (3.1) of the “Phenotypic plasticity of melanoma” section.

Response: The structure of paragraphs is now changed as suggested